# Taurine Prevents Angiotensin II-Induced Human Endocardial Endothelium Morphological Remodeling and the Increase in Cytosolic and Nuclear Calcium and ROS

**DOI:** 10.3390/nu16050745

**Published:** 2024-03-05

**Authors:** Danielle Jacques, Ghassan Bkaily

**Affiliations:** Department of Immunology and Cell Biology, Faculty of Medicine and Health Sciences, University of Sherbrooke, Sherbrooke, QC J1H 5N4, Canada; ghassan.bkaily@usherbrooke.ca

**Keywords:** human endocardial endothelial cells, angiotensin II, taurine, calcium overload, ROS, free radicals, hypertrophy

## Abstract

Endocardial endothelium (EE) is a layer of cells covering the cardiac cavities and modulates cardiomyocyte function. This cell type releases several cardioactive factors, including Angiotensin II (Ang II). This octopeptide is known to induce cardiac hypertrophy. However, whether this circulating factor also induces EE hypertrophy is not known. Taurine is known to prevent cardiac hypertrophy. Whether this endogenous antioxidant prevents the effect of Ang II on human EE (hEE) will be verified. Using quantitative fluorescent probe imaging for calcium and reactive oxygen species (ROS), our results show that Ang II induces (10^−7^ M, 48 h treatment) an increase in hEE cell (hEEC) volume and its nucleus. Pretreatment with 20 mM of taurine prevents morphological remodeling and increases intracellular calcium and ROS. These results suggest that the reported Ang II induces cardiac hypertrophy is associated with hEEC hypertrophy. This later effect is prevented by taurine by reducing intracellular calcium and ROS overloads. Thus, taurine could be an excellent tool for preventing Ang II-induced remodeling of hEECs.

## 1. Introduction

The heart cavities are covered with a monolayer of endothelial cells called the endocardial endothelium (EE) [1,2,3]. These cells are highly active. One important property of EE is its anti-coagulant and anti-thrombotic activities. This aspect is important for cardiomyocyte intracellular calcium homeostasis. The latter level is essential for the synthesis and secretion of vasodilatory factors such as nitric oxide (NO) and prostacyclin [4,5]. On the other hand, the EE secretes several vasoconstrictor cardioactive substances, including endothelin-1 (ET-1), prostaglandins, and several components of the renin–angiotensin system (RAS), such as Angiotensin II (Ang II) [3,5,6]. They also possess receptors at the plasma and nuclear membranes for ET-1, NPY, and Ang II [3,5,6]. Ang II acts via its receptors at the surface and nuclear membranes of EE cells (EECs), inducing an elevation of intracellular calcium concentrations ([Ca^2+^]_i_). This elevation in [Ca^2+^]_i_ may, in turn, induce remodeling of EE [6], hence its importance at the cardiac level. Moreover, a balance between the various factors secreted by EE is important for maintaining the intracellular ionic homeostasis and integrity of the endocardial endothelium.

Both EECs and vascular endothelial cells (VECs) regulate cardiac function [1]. However, although EECs and VECs have some similarities, there are significant differences [7], more particularly at the level of secretion of different circulating cardiovascular active factors [8]. The contribution of VECs to cardiac performance is due, at least in part, to the control of coronary blood flow supply to the myocardium [8]. However, the contribution of EECs to cardiac function is due to direct action on cardiomyocytes via the release and tuning of factors that affect not only cardiac function [1]. Thus, their close proximity to the adjacent cardiomyocytes affects the whole myocardium function.

Several studies have reported the presence of components of the renin–angiotensin II system (RAS) in endothelial cells, suggesting a prominent role for these cells in Ang II synthesis and secretion [9,10]. The presence of renin and its mRNA in ECs was reported, and their capacity to synthesize and secrete Ang II [11] was demonstrated. Angiotensin II converting enzyme (ACE) has also been identified in the endothelium’s luminal region at the cells’ plasma membrane [12]. In addition, ACE activity has been demonstrated in several endothelial cell preparations [11].

EECs release cardioactive factors such as Ang II in the extracellular matrix between these cells and ventricular cardiomyocytes [3]. Thus, they contribute to regulating cardiomyocyte function [3,7]. Evidence for the involvement of Ang II in cardiac hypertrophy is abundant [13,14,15]. This octapeptide, generated as part of the renin–angiotensin system, has been implicated in the pathophysiology of many cardiovascular diseases, such as peripheral artery disease, heart failure, hypertension, and coronary artery disease [16]. This circulating factor has been shown to induce hypertrophy of several cell types; however, nothing is known about the effect of Ang II on EEC hypertrophy, particularly those of human origin. It is reported that Ang II-induced cardiomyocyte hypertrophy is mediated via activation of intracellular signaling, particularly increasing ROS levels or decreasing anti-ROS production [17]. The increase in intracellular ROS by Ang II was reported to be mediated via the activation of c-Src causing an increase in NOX activity [17] and, more particularly, the Ca^2+^-dependent NOX5. In addition, taurine, a nonessential amino acid, is a well-known endogenous antioxidant and was recently reported to prevent the development of hypertrophy associated with heart failure and early death [18,19]. However, verifying whether taurine prevents Ang II’s effect on EECs via reducing Ang II-induced cell hypertrophy associated with increased intracellular ROS awaits verification.

Taurine is a well-known nonessential amino acid [19,20] that is very important for cell function. This circulating factor is produced mainly by the brain and the heart. Nutrients, particularly seafood [21,22,23,24,25,26], ensure its supplementation. Taurine’s beneficial effects are numerous, particularly at the cardiovascular level. Taurine was reported to be a cardiac anti-hypertrophic factor in vivo as well as in vitro [19,27]. Its overall beneficial effects were attributed in part to its anti-ROS properties [19,28] and in preventing intracellular calcium overload [3,19,27].

We took into consideration the following: 1—Ang II induces cardiac hypertrophy associated with an increase in cardiomyocyte intracellular Ca^2+^ and ROS [15]; 2—the first abnormal remodeling takes place at the EE during the development of hereditary cardiomyopathy, and it contributes to cardiovascular dysfunction [21]; 3—endothelial cell dysfunction is a complex mechanism [29] implicating ROS generation and overload that is also activated by Ang II [22,23,24,25]. Thus, in the present study, we wanted to verify whether Ang II induces hypertrophy of human EECs and whether this effect can be prevented by treatment with taurine.

## 2. Materials and Methods

### 2.1. Isolation and Culturing of Human Endocardial Endothelial Cells

The procedure for isolating and culturing hEECs was described previously. The procedures were performed in compliance with the institutional review committee’s requirements for using the human tissues of donors. In brief, hEECs were separated from the right ventricle of human fetal donor hearts. The right ventricle is open and cleaned from blood. For isolating the hEECs, the cardiac cavities were exposed to trypsin and then washed with the physiological ionic culture medium, the M199 solution containing 5% fetal bovine serum (*v*/*v*) (Life Technologies, Burlington, ON, Canada). The hEECs were gently isolated using a scalpel. Then, the cells were centrifuged at 200× *g* and added in the culture medium. As reported previously, isolated hEECs were cultured in Petri dishes. At confluence, hEECs were again isolated and recultured on glass coverslips placed in the culture dishes. The purity of cultured hEECs is verified using markers of these cell types.

### 2.2. Confocal Microscopy

As previously described, hEECs are studied using quantitative 3D confocal microscopy of a Bio-Rad system [26]. In summary, the laser line (9.0 mV) is directed to hEEC and is filtered to prevent photobleaching of the fluorescent dye. The confocal settings were kept unchanged in all the image recordings. The size between the cell sections (16–20 sections/cell depending on the value of the z line) is kept near zero to construct the real image of the cell in 3D. As reported previously, the nucleus is labeled with the fluorescent probe of nucleic acids, syto-11 (Molecular Probes, Eugene, OR, USA) [26]. Real 3D images are analyzed using an ImageSpace Rix version 6.5 analyzing system. This program allows quantitative 3D images to be obtained by measuring the volume of the cell (expressed in μm^3^).

### 2.3. Determination of the Cell Volume

Recorded images are transferred to an analysis station of Silicon Graphics equipped with Imagespace 3D analysis and reconstruction software from Molecular Dynamics. Images of cell volume are obtained and calcium measurements are performed on real three-dimensional reconstructions. The nucleus region, marked with Syto 11, is isolated from the rest of the cell by lowering the intensity threshold to delineate the pixels in this space. This method enables us to create a true three-dimensional reconstruction of the nucleus alone or of the cell without the nucleus. This method allows us to measure the fluorescence intensity values of the volume of the cell and the Fluo-4/Ca^2+^ complex of the nuclear region and the cytosolic region separately, eliminating any contribution from the other compartment.

### 2.4. Loading with the Calcium and ROS Fluorescent Probes

The cell membrane permeable Ca^2+^ fluorescence dye Fluo-4/AM (Molecular Probes, Eugene, OR, USA) is used to load Fluo-4 into the cytoplasm and the nucleoplasm of hEECs, as reported in our previous published work [26]. The calcium fluorescent probe Fluo-4 is homogeneously distributed in the hEECs [26] and can be expressed in free calcium concentration using a calibration method described previously. For ROS imaging studies, cells are loaded, respectively with the ROS/probe, 6-carboxy-2′,7′-dichlorodihydrofluorescein diacetate (carboxy-H2DCF-DA; Life Technologies, Burlington, ON, Canada). The methods have been developed and described previously.

### 2.5. Statistical Analyses

In this work, intracellular calcium and ROS levels are expressed as means ± SEM. *n* represents the number of hEECs from a minimum of three different experiments. N is the number of human donors. Student’s *t*-test and one-way repeated measures ANOVA with post hoc *t*-test (*p* < 0.05) are used along with the Tukey–Kramer or the Newman–Keuls and analysis by the program Graph Pad Prism version 8.4.3 (686).

## 3. Results

### Effect of Ang II on Human Ventricular Endocardial Endothelial Cells

To verify if Ang II induces hypertrophy of hEECs, the cells were exposed for 48 h to 10^−7^ M of Ang II. Our group reported that this concentration increased intracellular free calcium in several types of cells. Then, the volume of the cells was assessed using the fluorescent probe syto-11 coupled to quantitative 3D confocal microscopy [3]. Figure 1 illustrates typical images of the effect of Ang II treatment on hEECs. Figure 1A,B show an apparent increase in the cell volume, including the nucleus following chronic treatment with Ang II (10^−7^ M, for 48 h) (Figure 1B) compared to control (Figure 1A). This apparent increase in hEECs volume by Ang II is highly significant at the whole cell (Figure 2A; *p* < 0.01), the cytosol (Figure 2B; *p* < 0.01) and the nucleus (Figure 2C, *p* < 0.01) levels. On the other hand, 48 h treatment with Ang II in the presence of 20 mM of taurine [19] prevents Ang II-induced increase in EEC apparent volume (Figure 1C). This effect of taurine was highly significant at the whole cell (*p* < 0.0001), the cytosol (*p* < 0.001), and the nucleus (*p* < 0.0001) levels.

In a second set of experiments, using the protocols described earlier, coupled to the calcium fluorescent probe Fluo-4, we examined if an increase in cytosolic and nuclear calcium accompanies the hypertrophy induced by a 48 h Ang II treatment. Figure 3 shows typical results. Figure 3A shows, as expected, a higher calcium level in the nucleoplasm than in the cytosol. Figure 3B shows that the hypertrophy induced by a 48 treatment with Ang II is associated with a very high increase in free calcium that is significant at the level of the whole cell (Figure 4A; *p* < 0.0001), the cytoplasm (Figure 4B; *p* < 0.01), and the nucleoplasm (Figure 4C; *p* < 0.001).

In the following experiments, we verified whether the prevention of Ang II-induced hypertrophy associated with increased intracellular calcium is prevented by treatment with taurine (20 mM). As seen in Figure 3C and Figure 4, treatment with taurine prevents Ang II-induced hypertrophy and the increase in the whole cell (Figure 4A; *p* < 0.01), cytosolic (Figure 4B; *p* < 0.05), and nuclear (Figure 4C; *p* < 0.05) calcium levels.

Using the same protocol described previously, we verified whether the Ang II-induced hEEC hypertrophy and intracellular calcium overload are associated with increased cytosolic and nuclear ROS overload. Figure 5 shows typical examples. As seen in this figure, intracellular ROS is distributed non-homogenously (Figure 5A). Figure 5B shows that Ang II-induced hEEC hypertrophy is accompanied by an apparent increase in ROS throughout the cells. As seen in Figure 5B,C, in the presence of Ang II, taurine apparently decreased ROS levels throughout the cell except for the nucleus. This ROS increase is statistically significant in the whole cell (Figure 6A; *p* < 0.001), the cytoplasm (Figure 6B; *p* < 0.001), and the nucleoplasm (Figure 6C; *p* < 0.05). In the presence of Ang II, treatment with 20 mM of taurine partially prevents the Ang II-induced increase in ROS at the whole cell level (Figure 6A; *p* < 0.05). However, it completely prevented the increase in cytoplasmic ROS level compared to the control ROS level (Figure 6B; ns) and did not affect nuclear ROS level (Figure 6C; ns).

## 4. Discussion

The dialogue between EECs and the cardiomyocytes occurs during the fetal formation of the heart. This continues during fetal cardiac development and cardiomyocytes differentiation [8]. Signaling, including secretion of EECs contributes to the development and maturation of ventricular trabecula that characterize the heart ventricles. Several reports have demonstrated that the contribution of EECs to heart function continues during an individual’s life. This is due to the fact that EECs are essential for the survival of postnatal cardiomyocytes and for the integrity of the heart [30]. In addition, EECs influence myocardial performance [31]. Brutsaert et al. were the first to report that in the absence of EECs, the contraction of cat cardiac papillary muscles resulted in a shortening of twitch contractions and a decrease in peak twitch force [8,32]. Similar work was reported by Fort et al.’s group using the whole heart [33]. The contractile performance of the heart depends primarily on the level of increase in transient intracellular Ca^2+^ [8].

Furthermore, EECs modulate the affinity of contractile proteins to an intracellular elevation of Ca^2+^ [8] in chronic high-circulating Ang II. Two mechanisms have been proposed to explain the modulatory effect of Ang II on EE and its regulation of myocardial performance: 1—upon stimulation with Ang II, EECs secrete several factors that alter the contractile state of the subjacent myocytes (excitation–secretion–contraction coupling) [8]; 2—the EE may act as a physicochemical barrier which controls the ionic constitution of the interstitial milieu surrounding the cardiomyocytes, and consequently, it modulates their performance (EE as a blood–heart barrier) [2,8,32]. Thus, modulation of EE function by Ang II will affect its excitation–secretion coupling and, consequently, will modulate the adjacent cardiomyocytes’ excitation–contraction coupling and EE morphological remodeling, leading to cardiac hypertrophy. Our results, using quantitative 3D confocal microscopy, demonstrate that long-term treatment with Ang II (48 h) increased the volume of hEECs. This increase is associated with an increase in the volume of the nucleus. The latter directly indicates that the increase in volume is indeed hypertrophy [34]. These findings demonstrate that Ang II-induced cardiac hypertrophy [13,14] is associated with hEEC hypertrophy. The latter will promote the release of hEECs growth factors such as ET-1 and Ang II, which contribute to the development of the well-known cardiac hypertrophy [13,14]. Ang II-induced hypertrophy could be partly due to the activation of AT1 and AT2 receptors. The latter promotes the Ang II-induced morphological remodeling of all cardiac cell types, including EECs. Our results also show that the Ang II-induced hypertrophy of hEECs is associated with increased intracellular calcium. Such an effect is similar to those effects reported in cardiomyocytes, vascular endothelial, and smooth muscle cells [3,35].

The increase in hEECs nuclear calcium by Ang II could be due to Ang II activation of the following: 1—calcium influx through the nuclear membranes ionic transporters; 2—release of calcium from endoplasmic IP_3_ sensitive pools [3]; and 3—release of calcium from nucleoplasmic reticulum IP_3_ [13]. Since endothelial cells generally possess mainly R-type calcium channels [15,23], the increase in cytosolic and nuclear calcium could also be due to the activation of plasma and nuclear membranes’ R-type calcium channels, as reported previously.

Our results also show that, as for other types of cells [3], Ang II-induced hEEC hypertrophy is also associated with increased cytosolic and nuclear ROS levels. This ROS increase occurs in several types of hypertrophies, including that induced by Ang II [13,14]. The increase in cytosolic and nuclear ROS could be due, at least in part, to activation of the calcium-depended NOX5.

Our results show that, as in hypertrophy associated with cardiomyopathy [19], taurine prevents Ang II-induced hEEC hypertrophy. This suggests that the reported preventive effect of taurine in hypertrophic cardiomyopathy [19] could be due at least in part to beneficial effects in the prevention of EEC hypertrophy. This should be verified in the future. In addition, the prevention of Ang II-induced hypertrophy and increase in intracellular free calcium could be due to the long-term effect of taurine in reducing calcium overload [18,19]. Our results also show that taurine prevents Ang II-induced increase in cytosolic ROS without affecting the increase in ROS at the nucleoplasmic level. These results suggest that ROS generation at the cytosolic level is differently generated in the cytosol when compared to the nucleoplasm. This could be due to differences in localization and density of NOXs (1–5) at the nuclear level compared to the cytosol. Since taurine prevents Ang II from inducing an increase in nuclear volume and calcium without affecting nuclear ROS levels in hEECs, this suggests that the latter does not contribute to the hypertrophy induced by Ang II.

Taurine supplementation also increases the proteomic and mRNA levels of protein kinase A-cAMP response element-binding protein (PKA-CREB) [36,37]. This transcription factor prevents cardiovascular cells’ morphological remodeling and explains, at least in part, taurine’s anti-trophic action on Ang II-induced EE hypertrophy. It is known that an increase in Ang II modulates the activity of the Na/Ca exchanger. This promotes calcium entry via this exchanger, accumulating calcium at both the cytosolic and nuclear levels [19,27,38]. It is possible that taurine indirectly blocks the activity of this exchanger via its inhibition of calcium-dependent kinases such as protein kinase C (PKC) [39,40]. It is unlikely that taurine blocks the activity of Ang II-activated receptors. On the other hand, as indicated above, it is likely that, at least in part, taurine’s anti-hypertrophic effect in EE is due to its inhibition of Ang II signalization cascades, such as those responsible for activation of the PKA-CREB and intracellular ROS generation pathways [17]. Thus, the prevention of Ang II-induced hypertrophy in hEECs by taurine could be mainly due to taurine blunting the increase in oxidative stress induced by Ang II [17,19].

We have to mention that the nucleus also acts as a cell within a cell. The nuclear membranes possess ACE and thus can generate Ang II, which will activate its receptors AT1 and AT2 at the nuclear membranes’ levels. The activation of these nuclear membrane receptors by Ang II-induced calcium influx through the nuclear envelope membranes is independent of the increase in cytosolic Ca^2+^. The internalization of Ang II promotes the activation of the nuclear membranes’ AT1 and AT2 receptors. The activation of these receptors stimulates Ca^2+^ influx through the nuclear membranes’ R-type Ca^2+^ channels. The latter may explain, at least in part, the increase in nuclear Ca^2+^ by Ang II in EECs. Since taurine influx takes place via its plasma membrane symporters, which increases its intracellular level, the latter may directly act at the nuclear level, thus preventing remodeling of the nucleus by Ang II.

## 5. Conclusions

In conclusion, our results clearly show that Ang II does induce hypertrophy of EECs, and these types of cells may contribute to the overall effect of Ang II in the heart. In addition, taurine’s prevention of Ang II-induced EEC hypertrophy demonstrates that the preventive effect of this nonessential amino acid on cardiac hypertrophy in hereditary cardiomyopathy [18,19] is due, at least in part, to its prevention of EEC hypertrophy.

## Figures and Tables

**Figure 1 nutrients-16-00745-f001:**
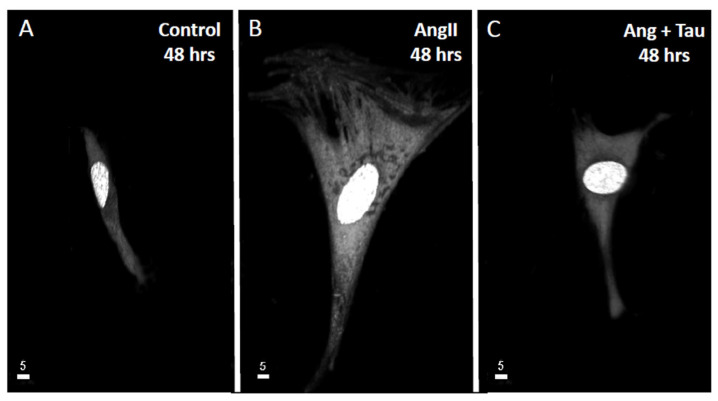
An example of real 3D top view confocal microscopy images showing the apparent increase in hEECs volume and its prevention by taurine. Images (**A**–**C**) show syto-11 labeled cells and their nucleus: (**A**) in the absence of angiotensin II (Ang II), (**B**) in the presence 48 h of treatment with 10^−7^ M of Ang II, and (**C**) in the presence of Ang II and taurine (Tau, 20 mM). The calibration of the white bar is in μm. Images (**A**–**C**) represent different cells.

**Figure 2 nutrients-16-00745-f002:**
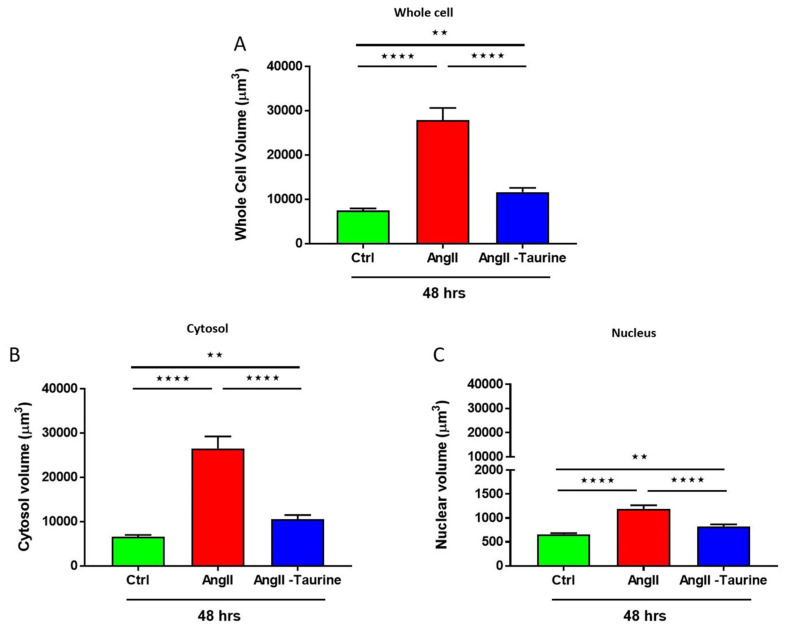
In human endocardial endothelial cells, treatment for 48 h with 10^−7^ M Ang II-induced an increase in the whole cell volume ((**A**), red color), the cytosol ((**B**), red color), and the nucleus ((**C**), red color). Treatment with taurine for 48 h induced a partial but significant prevention of Ang II-induced increase in cell volume at the whole cell ((**A**), blue color), the cytosolic ((**B**), blue color), and the nucleoplasmic ((**C**), blue color) levels. The results are presented as mean ± SEM; n is the number of cells from five donors (N). ** *p* < 0.01; **** *p* < 0.0001. Results are presented in µm^3^.

**Figure 3 nutrients-16-00745-f003:**
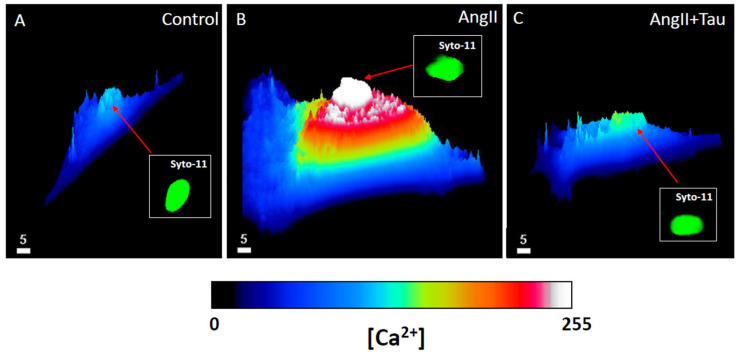
An example of a quantitative 3D fishnet plot representation showing the apparent cytoplasmic and nucleoplasmic free Ca^2+^ levels in hEECs. Image (**A**) represents hVSMCs Ca^2+^ levels and distribution in a normal control medium. Image (**B**) represents hEECs Ca^2+^ levels and distribution in the presence of Ang II (10^−7^ M) for 48 h. Image (**C**) represents hEECs Ca^2+^ levels and distribution in the presence of 48 h of treatment with Ang II (10^−7^ M) and taurine (20 mM). Insert panels (**A**–**C**) in green color (color has no measurable meaning) represent syto-11 labeled nucleus in the absence of angiotensin II (Ang II) (**A**), in the presence 48 h treatment with Ang II (**B**), and in the presence of Ang II and taurine (**C**). The white bar scale is in μm. Images (**A**–**C**) represent different cells. The color calibration bar represents the free Ca^2+^ fluorescence level from 0 (black color) to 255 (white color).

**Figure 4 nutrients-16-00745-f004:**
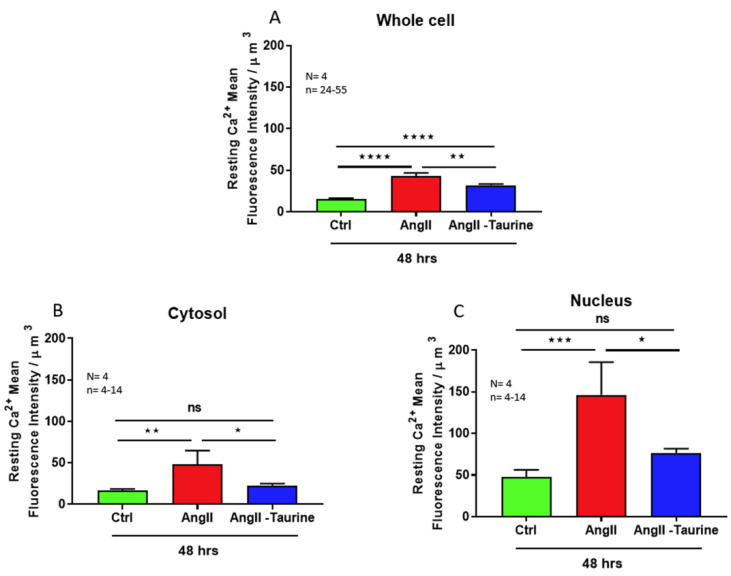
In hECCs, treatment for 48 h with Ang II (10^−7^ M) elevated intracellular free Ca^2+^ in the whole cell ((**A**), red color), the cytoplasm ((**B**), red color), and the nucleoplasm ((**C**), red color). Treatment with taurine for 48 h induced partial but statistically significative prevention of Ang II-induced increase in Ca^2+^ free levels in the whole cell ((**A**), blue color), and completely prevented the Ang II-induced increase in Ca^2+^ free levels in the cytoplasm ((**B**), blue color) and the nucleoplasm ((**C**), blue color). The results are expressed as mean ± SEM, and n represents the number of cells from five donors (N). * *p* < 0.05; ** *p* < 0.01; *** *p* < 0.001; **** *p* < 0.0001; ns: not significant. Results are presented per µm^3^ of the cell volume.

**Figure 5 nutrients-16-00745-f005:**
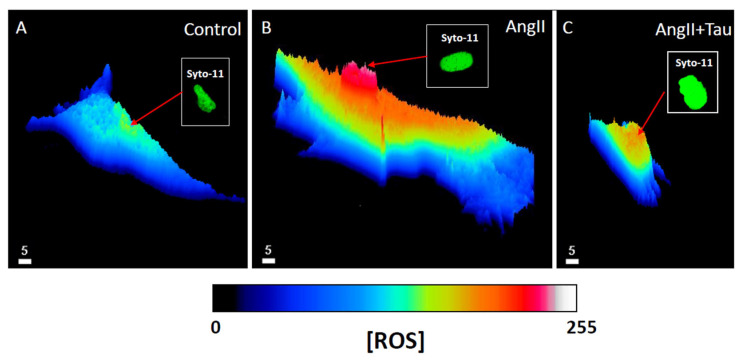
An example of quantitative 3D confocal microscopy images (3D fishnet plot representation) showing the apparent cytoplasmic and nucleoplasmic ROS levels in hEECs. Image (**A**) represents hEECs ROS levels and distribution in a normal control medium. Image (**B**) represents hEECs ROS levels and distribution in the presence of Ang II (10^−7^ M) for 48 h. Image (**C**) represents hEECs ROS levels and distribution in the presence of 48 h of treatment with Ang II (10^−7^ M) and taurine (20 mM). Insert panels (**A**–**C**) in green color (color has no measurable meaning) represent syto-11-labeled nucleus in the absence of angiotensin II (Ang II) (**A**), in the presence of 48 h treatment with Ang II (**B**), and in the presence of Ang II and taurine (**C**). The white bar calibration is in μm, and (**A**–**C**) are from different cells. The color calibration bar is ROS fluorescence levels ranging from 0 (black color) to 255 (white).

**Figure 6 nutrients-16-00745-f006:**
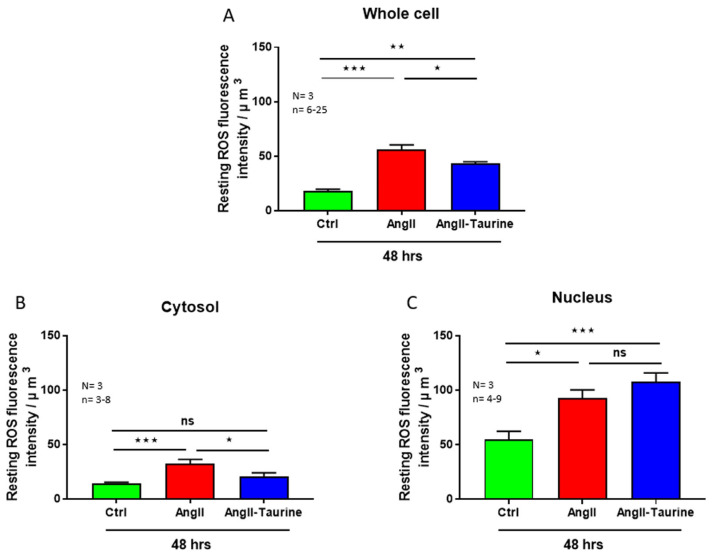
In hEECs, treatment for 48 h with Ang II (10^−7^ M)-induced elevation in ROS levels in the whole cell ((**A**), red color), the cytoplasm ((**B**), red color), and the nucleoplasm ((**C**), red color). Treatment with taurine for 48 h partially but significantly prevented the Ang II-induced increase in ROS levels in the whole cell ((**A**), blue color), and in the cytoplasm ((**B**), blue color), but not in the nucleus ((**C**), blue color). The results are expressed as mean ± SEM, and n is the number of cells from three donors (N). * *p* < 0.05; ** *p* < 0.01; *** *p* < 0.001; ns: not significant. Results are presented per µm^3^ of the cell volume.

## Data Availability

Data are contained within the article.

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
