# Peer review of "Taurine Prevents Angiotensin II-Induced Human Endocardial Endothelium Morphological Remodeling and the Increase in Cytosolic and Nuclear Calcium and ROS"

_nutrients, 2024, doi:10.3390/nu16050745_

Round 1

Reviewer 1 Report

Comments and Suggestions for Authors

 First, I have some formal requirements that I address to you in order to correct them.

1. What does repetition 24 represent in the context of 10-7M, 24 24-hour treatment line 17.

2. To space in the context This increase in [Ca2+]i may line 37.

3. What is the meaning of M199 solution. It is correct line 92.

4. Then, the cells were centrifuged for 10 min at 200g and resuspended in the culture medium. In this context, 200g is correct line 93.

Secondly, I will ask other questions.

1.. Why does reference [1] appear the first time after reference [9] in the text.

2. To correct the references in full according to the instructions for the authors.

 3. No bibliographic reference contains the necessary http link and according to the instructions.

for example:

  1. Rauf, A.; Imranb, M.; Patel, S. Mangiferin: A phytochemical with panacea potential. Biomed. Pharmacother. 201796, 1562–1564. http

2. How do you explain the essential element of originality, to be convincing. As you specify in the manuscript reference 59.

"The present results indicate that taurine is an effective inhibitor of angiotensin II action. It is plausible that the beneficial effect of taurine in the treatment of heart failure could relate to its suppression of angiotensin II-mediated cellular responses.''

Azuma, M.; Takahashi, K.; Fukuda, T.; Ohyabu, Y.; Yamamoto, I.; Kim, S.; Iwao, H.; Schaffer, S. W.; Azuma, J., Taurine attenuates hypertrophy induced by angiotensin II in cultured neonatal rat cardiac myocytes. Eur J Pharmacol 2000, 403, (3), 486 181-188.

Author Response

Answers to reviewer 1: We thank the reviewer for his/her comments and suggestions. We did revise the manuscript to your suggestions. All is highlighted in red.

  • What does repetition 24 represent in the context of 10-7 M, 24 24-hour treatment line 17.

Answer: We are sorry about the mistake, and now it is corrected to 48-hour.

  • To spacein the context This increase in [Ca2+]i may line 37.

Answer: As suggested, we define [Ca2+]i.

  • What is the meaning of M199 solution. It is correct line 92.

Answer: As suggested, we define the culture medium M199.

  • Then, the cells were centrifuged for 10 min at 200g and resuspended in the culture medium. In this context, 200g is correct line 93.

Answer: Yes, it is. Thank you for your comment

Secondly, I will ask other questions.

  • Why does reference [1] appear the first time after reference [9] in the text.

Answer: Thank you for pointing out this matter and now it is corrected.

  • To correct the references in full according to the instructions for the authors.

Answer: This was done according to Endnotes of Nutrients. The journal will make any changes as usual. 

  • No bibliographic reference contains the necessary http link and according to the instructions

Answer: This was done according to Endnotes of Nutrients. Any changes will be made by the journal as usual.

  • How do you explain the essential element of originality, to be convincing. As you specify in the manuscript reference 59.

Answer: We did revise this aspect in the introduction and discussion sections.

  • "The present results indicate that taurine is an effective inhibitor of angiotensin II action. It is plausible that the beneficial effect of taurine in the treatment of heart failure could relate to its suppression of angiotensin II-mediated cellular responses.''

Answer: We did revise this aspect in the revised manuscript.

  • Azuma, M.; Takahashi, K.; Fukuda, T.; Ohyabu, Y.; Yamamoto, I.; Kim, S.; Iwao, H.; Schaffer, S. W.; Azuma, J., Taurine attenuates hypertrophy induced by angiotensin II in cultured neonatal rat cardiac myocytes. Eur J Pharmacol 2000, 403, (3), 486 181-188.

Answer: We made it clear that the originality of our work is to show that the hypertrophy induced by Ang II is not limited to cardiomyocytes but also to endocardial endothelium. As cited in the revised manuscript, this EEC hypertrophy would modulate the excitation-secretion coupling of EECs and release hypertrophic factors such as endothelin-1 and neuropeptide Y. This will be verified in the future. In addition, our work is done in human EECs and shows clearly that taurine's effect is mainly mediated via decreasing intracellular ROS elevation induced by Ang II.

Reviewer 2 Report

Comments and Suggestions for Authors

Unlike humans, cats need taurine as an essential amino acid (one of the reasons they are obligate carnivores and cannot be forced to become vegan). A significant symptom of feline taurine deficiency is dilated cardiomyopathy. Other than primates, the cat-human comparison is one of the closest you can get, in terms of genome organization. Dogs and mice, by contrast, have chromosomes that have been reshuffled over their respective evolutionary histories, making them more complicated to use as genetic analogues for our species. Given the effects of taurine deficiency in cats, It is to some extent therefore not a surprise that taurine has significant effects on human endothelial cells.

There is already a lot of information known about the effects of taurine on cells (e.g. https://www.ncbi.nlm.nih.gov/pmc/articles/PMC5933890/); it is known to increase intracellular calcium (https://link.springer.com/chapter/10.1007/978-1-4615-5763-0_20 ; and cell size (https://pubmed.ncbi.nlm.nih.gov/9871493/); and to modulate oxidative stress in mitochondria (https://academic.oup.com/abbs/article/45/5/359/1303) and there have been other similar studies by confocal microscopy.

There must needs to be some reason why this study has been completed: It is therefore good that it has been  carried out in human endothelial cells with a test of the effect of taurine against angiotensin. Overall, the experimental methods seem appropriate. However, results presentation could be improved because bar and SEM plots are a very poor way to represent the data. It would be more informative to use a box and whisker plot for figures 2, 4 & 6.

The discussion is comprehensive and clear.

Author Response

Answer to reviewer 2: We thank the reviewer for his/her comments and suggestions. We did revise the manuscript to your suggestions. All is highlighted in red.

  • Unlike humans, cats need taurine as an essential amino acid (one of the reasons they are obligate carnivores and cannot be forced to become vegan). A significant symptom of feline taurine deficiency is dilated cardiomyopathy. Besides primates, the cat-human comparison is one of the closest you can get to genome organization. Dogs and mice, by contrast, have chromosomes that have been reshuffled over their respective evolutionary histories, making them more complicated to use as genetic analogues for our species. Given the effects of taurine deficiency in cats, It is to some extent therefore not a surprise that taurine has significant effects on human endothelial cells.

Answer: Not only cats have a taurine deficiency but also humans because the body does not sufficiently reduce taurine and depends on diet supplements, mostly in seafood. This is well-reviewed by one of our recent review papers dealing with taurine. We must mention that taurine's preventive effect on long-term Ang II exposure is not apparent in vascular endothelial cells and is absent in EECs. Therefore, our work is highly original. 

  • There is already a lot of information known about the effects of taurine on cells (e.g. https://www.ncbi.nlm.nih.gov/pmc/articles/PMC5933890/); it is known to increase intracellular calcium (https://link.springer.com/chapter/10.1007/978-1-4615-5763-0_20 ; and cell size (https://pubmed.ncbi.nlm.nih.gov/9871493/); and to modulate oxidative stress in mitochondria (https://academic.oup.com/abbs/article/45/5/359/1303) and there have been other similar studies by confocal microscopy.

Answer: We thank the reviewer for his/her comment. We were the first to show that short-term exposure to taurine increases intracellular calcium. However, we later showed that taurine's long-term effect is to decrease intracellular calcium overload. We looked at the paper of the group of Schaeffer that was suggested by the reviewer. Unfortunately, we cannot access the full paper because this manuscript is old. Looking at the abstract, this paper talks about decreasing intracellular taurine by blocking the sodium-taurine symporter, which reduces cell size and not necessarily cell volume due to the effect of taurine in cell swelling and not cell hypertrophy. Thus, this paper has nothing to do with our EEC hypertrophy.        

  • There must be some reason why this study has been completed: It is, therefore, good that it has been carried out in human endothelial cells with a test of the effect of taurine against angiotensin. Overall, the experimental methods seem appropriate. However, results presentation could be improved because bar and SEM plots are a very poor way to represent the data. Using a box and whisker plot for figures 2, 4 & 6 would be more informative.

Answer: As suggested, we clarified the novelty of our results in the revised manuscript. In all our published papers, we prefer SEM plots instead of box and whisker plots. The two latter plots make the figure busier and do not teach us anything. However, SEM does. Whatever the type of plots used, the significance is always the same. 

Reviewer 3 Report

Comments and Suggestions for Authors

This work presents valuable research with high originality and scientific soundness. However, there are areas where improvements could enhance the overall impact of your work. The quality of the presentation can be refined for clarity and engagement; consider simplifying complex language to make it more accessible to a broader audience. While the content is significant, increasing its appeal to readers outside your immediate field could broaden its impact. This could be achieved by highlighting the broader implications of your findings in the introduction and conclusion sections. Additionally, authors should mention through which mechanism Ang II induces hypertrophy, and through which mechanism Taurine reverses the hypertrophy. This would not only clarify the underlying biological processes but also set the stage for future research, potentially leading to a follow-up paper. 

Author Response

Answer to reviewer 3: We thank the reviewer for his/her comments and suggestions. We did revise the manuscript to your suggestions. All is highlighted in red.

  • This work presents valuable research with high originality and scientific soundness. However, there are areas where improvements could enhance the overall impact of your work. The quality of the presentation can be refined for clarity and engagement; consider simplifying complex language to make it more accessible to a broader audience. While the content is significant, increasing its appeal to readers outside your immediate field could broaden its impact. This could be achieved by highlighting the broader implications of your findings in the introduction and conclusion sections. Additionally, the authors should mention which mechanism Ang II induces hypertrophy and which mechanism Taurine reverses the hypertrophy. This would clarify the underlying biological processes and set the stage for future research, potentially leading to a follow-up paper.

Answer: We simplified the manuscript's language as much as we could. We also highlighted the findings in the introduction and conclusion sections, as shown in the highlighted red text. We also proposed mechanisms that could be implicated in Ang II induced hypertrophy and more particularly activation of NOX5 as well as taurine reduction of Ang II induced increase in ROS. 

Round 2

Reviewer 1 Report

Comments and Suggestions for Authors

I appreciate your correction. But I don't think you understood the numbering of the references in the text.

 It would be correct [1] Kuruvilla, L. in the text to appear before [2]  Brutsaert, D. line 26

Reference [1] appears unchanged after 9, before 10. line 38

As in the previous form (18 is repeated with 31), and now why is the reference 18 and 32 repeated. They are similar.

[18] Jacques, D. Bkaily, G., Cardiovascular physiopathology of angiotensin II and its plasma and nuclear envelop membranes'  receptorsEds N.S. Dhalla et al., Eds N.S. Dhalla et al., In The renin Angiotensin system in cardiovascular disease. Advances in Biochemistry in Health and Disease, N.S. Dhalla et al., Ed. Springer Nature Switzerland, 2023; pp 63-80.

[32] Jacques D., Bkaily G., Cardiovascular Physiopathology of Angiotensin II and Its Plasma and Nuclear Envelop Membranes' Receptors. In The Renin Angiotensin System in Cardiovascular Disease, Advances in Biochemistry in Health and Disease, al., N. S. D. e., Ed. Springer Nature Switzerland AG: 2023; Vol. 24, pp 63-80.

You didn't link to every reference.

This link is missing, it should be added to each bibliographic reference. [Google Scholar] [CrossRef]

Example:

  1. Djuricic, I.; Calder, P.C. Beneficial Outcomes of Omega-6 and Omega-3 Polyunsaturated Fatty Acids on Human Health: An Update for 2021. Nutrients 202113, 2421. [Google Scholar] [CrossRef]

Author Response

We thank the reviewer for his/her helpful comments. We apologize for the mistake concerning the references. For a reason that we still ignore, upon loading the revised manuscript, our corrections for the references disappear without our knowledge. The revisions are made according to the reviewer's comments. However, endnotes do not cite Crossref and Google Scholar. The journal usually does this once the proof is accepted.